# Treatment Beliefs and Practices towards Low Back Pain among Teachers in Asir Region, Saudi Arabia—A Cross-Sectional Study

**DOI:** 10.3390/healthcare11030383

**Published:** 2023-01-29

**Authors:** Abdullah Raizah, Faris Alzahrani, Bandar Albarqi, Ibrahim Abusaq, Hashim Alqarni, Ibraheem Alyami, Irshad Ahmad, Ravi Shankar Reddy

**Affiliations:** 1Department of Orthopaedic Surgery, King Khalid University, Abha 61413, Saudi Arabia; 2The Joint Program of Public Health and Preventive Medicine, Abha 62527, Saudi Arabia; 3Department of Pediatric Surgery, University Hospital Estaing, 63000 Clermont Ferrand, France; 4Ministry of Health, Abha 62523, Saudi Arabia; 5Department of Medical Rehabilitation Sciences, College of Applied Medical Sciences, King Khalid University, Abha 61413, Saudi Arabia

**Keywords:** treatment beliefs, low back pain, teachers

## Abstract

The prevalence of low back pain (LBP) is increasing exponentially, with this public health issue affecting over 70% of the population. However, sedentary careers exacerbate the problem further, with professions such as teaching disproportionately affected. In addition, the general population does not seek interventions from medical professionals for LBP; instead, they opt to manage their pain with over-the-counter medications, such as sedatives. The purpose of this study was to explore practices and beliefs related to back pain treatment among schoolteachers in the Asir region. This cross-sectional study included a sample of 312 teachers from the Asir region, with data collected regarding the prevalence of back pain, management approaches, and beliefs surrounding medical interventions. Chi-square or exact tests defined the association between variables, with significance determined at *p* < 0.05. Our results revealed that 67.3% of Saudi Arabian teachers experienced LBP within the last two months, with a continuous condition representing 36.7% of cases. This study identified several different treatment modalities the participants use to manage their pain, including sedatives, best rest, kaiy (traditional cautery), and local adhesives, with sedatives being the most employed method. It also highlighted that increased daily working hours and total days worked significantly increased the prevalence of LBP (*p* < 0.05). Although a considerable number of the included teachers were highly educated, with some teaching experience, they had a poor level of awareness and an incorrect attitude towards pain management. Enhanced efforts should be made to improve teachers’ awareness regarding back pain causes and management methods.

## 1. Introduction

Low back pain (LBP) represents a public health issue affecting 70 to 80% of the population. This condition may also influence the patient’s social and economic status [1]. Moreover, most LBP cases do not have a specific cause [2]. This condition is self-limiting and mostly treated at a primary care facility within 12 months of the initial diagnosis of LBP. It has also been noted that three-quarters of patients have persistent symptoms such as pain and disability [1,3].

According to a single study that was done in Al-Qassim, it was found that the prevalence of LBP in the Saudi Arabian general population is 18.8% [4]. An additional study also showed that females experienced worse symptoms; hence, they made more complaints [5]. Moreover, other studies have been done on different categories of people in different cities in Saudi Arabia. These investigations are compared to other studies conducted worldwide [6].

Schoolteachers represent an occupational group among which there appears to be a high prevalence of MSD [7]. Musculoskeletal complaints, especially of the lower back, neck, and shoulders, are common among teachers. MSD is one of the most common and expensive occupational health problems in developed and developing countries. Epidemiological studies have also demonstrated that gender, age, length of employment, and awkward posture are associated with higher MSD prevalence rates among teachers [8]. MSD is one of the leading causes of sick leave, absenteeism, and ill-health retirement among schoolteachers [7,9,10]. Harcombe et al. [11] confirmed that low back pain is a common problem in both heavy and light manual workers.

Recently, Hong Kong teachers reported a higher prevalence of neck (68.9%), shoulder (73.4%), and low back pain (59.2%) in the past 30 days. It is worth noting that the sample of Hong Kong teachers showed a significantly higher prevalence in all single musculoskeletal complaints than the Norway sample in one study [12]. Al-Rowayeh et al. [13] performed a study with 308 teachers in Kuwait to see the prevalence and related reasons for back pain in teachers; its finding showed that 68.5% of teachers reported back pain.

Despite the high prevalence of LBP problems in general and specifically in teachers, the current literature fails to discuss this health problem among teachers in Saudi Arabia. Therefore, the current study aims to explore the practices around and beliefs about back pain treatment among schoolteachers in the Asir Region.

## 2. Materials and Methods

A sample of 312 teachers from different governmental schools in the Asir region was included. A list of all available schools was obtained from the ministry of education website in the Asir region. All available teachers present during the visit time were invited to participate. Data were collected after approval using a pre-structured questionnaire following an extensive literature review. The questionnaire validity was confirmed by three experts who reviewed the questions in detail, and any suggestions for improvement were implemented after consensus. Reliability ranged from 0.69 for beliefs regarding physiotherapy to 0.74 for analgesic effectiveness beliefs.

### Statistical Analysis

After collection, the data were refined, coded, and fed to IBM SPSS version 21. Descriptive statistics were used to show frequencies and percentages for categorical variables, whilst the mean was used to display scale variables. Chi-square or exact tests were used to test for an association between a history of having back pain and different sample characteristics and beliefs. All tests were two-tailed, with a *p* value less than or equal to 0.05 considered significant.

## 3. Results

The study included 312 teachers from different schools, of which 87.8% were males. The age of participants ranged from 18 to 59 years, with a mean value of 39.2 years. Most participants were of middle age (84.9%, 30–50 years). Most of the included teachers (92.6%) were Saudi, and more than 70% were from Abha or Khamis Muchait cities. Regarding educational level, 53.9% of the sampled staff had bachelor’s degrees, 29.9% were postgraduates, and 16.2% had general diplomas. Considering marital status, 97.7% of the teachers were single, and 82.1% were smokers (Table 1).

Regarding work data, 89.7% of the interviewed staff were teachers, while the others worked in administrative positions (Table 2). Most sampled teachers (94.3%) worked five days during the week, with only 4.3% working every day. In addition, over 50% of the teachers worked for seven hours daily, 24% for six hours daily, and only 1.4% for four hours daily.

By asking the interviewed teachers about back pain, 67.3% of participants revealed that they had experienced back pain within the last two months, which was noted as continuous in 36.7%. The onset of pain during the last week was recorded among 55.7% of those with a back pain history. On asking about the consequences of having back pain, 38.1% of those who had the pain highlighted that it affected their performance, 29.5% stated that pain affected their social relations, and only 19.5% recorded previous absenteeism due to back pain (Figure 1).

Regarding the actions taken to overcome their back pain, 46.5% of participants used sedatives as pain killers, followed by bed rest (33.7%), kaiy (31.7%), and local adhesives (28.8%). In comparison, 9.9% of them visited a primary healthcare center for medical advice and 3.5% visited an ER department (Figure 2).

Table 3 shows the attitude of the participants regarding pain-relieving methods. As for sedatives, 7.4% of the sample agreed on their effectiveness as a pain relief method, whilst 27.6% of them stated that they were useless. On the other hand, 17% of participants feared using sedatives, whilst 40.1% found them to be effective pain killers. As for doing exercises, 8% of participants agreed on its effectiveness as a pain relief method, 37.5% found it unhelpful, 30.1% of participants were afraid of doing exercise, and 15.1% agreed on its role as a pain killer. Considering physiotherapy, 6.7% of the sample agreed on its role in pain-relieving, 40.4% recorded it as a useless intervention, 28.5% were afraid of physiotherapy, and 9.6% found it to be an effective pain killer. Finally, considering the body piercing method, 16.1% of participants agreed on its role as a pain relief method, 19.9% found it provided no pain relief, 11.9% of participants were afraid of using this method, and 22.8% said it is an effective pain killer.

When evaluating the participant’s history of back pain (Table 4), it was evident that 76.7% of teachers below the age of 30 years had back pain history compared to 70.6% of those who were above 50 years old with no statistical significance (*p* > 0.05). Considering gender, 76.3% of the females recorded pain history compared to 66.1% of males (*p* > 0.05). In addition, 65% of the teachers had previously complained of back pain compared to 71.4% of those in administrative positions, but with no significant difference (*p* > 0.05). Regarding weekly working days, all those who worked every day had a history of back pain compared to 50% of those who worked six days and 72.9% of those who worked five days with recorded statistical significance (*p* < 0.05). Considering daily working hours, 74.8% of those who worked for 7 h daily had back pain compared to 66.7% of those who worked for 4 h daily and 62.2% of those who worked for 5 h (*p* < 0.05).

## 4. Discussion

Our results found that 67.3% of teachers in Saudi Arabia experience LBP. Of our participants, 67.3% had experienced back pain within the last two months, with a continuous condition in 36.7% of cases. However, the pain experienced by participants is not the only implication of LBP. Our findings also suggest that back pain impacts job performance, social relations, and absenteeism, with a prevalence of 38.1%, 29.5%, and 19.5% recorded in the participants, respectively. Moreover, this study also identified several treatment modalities our participants implemented to manage their pain, including sedatives, bed rest, kaiy, and local adhesives.

### 4.1. Prevalence of LBP

Of our participants, 67.3% had experienced back pain within the last two months.. Aldukhayel et al. [14] conducted a cross-sectional study assessing the prevalence, patterns, and risk factors of musculoskeletal pain disorders, including LBP, amongst Saudi Arabian teachers. The findings of this study were slightly higher than ours, with a prevalence of 74.4% being recorded. Another study by Abdulmonem et al. [15] noted a prevalence of lower back pain of 38.1% among female teaching staff in Saudi Arabia. However, the discrepancies between these findings and our study can be attributed to Abdulmonem et al. [15] solely investigating the prevalence of severe LBP in this population. A more representative figure, in line with our aims, was reported in a cross-sectional study by Abdel-Salam et al. [16] that recorded a prevalence of 68.4% for LBP in female teachers in Saudi Arabia. A study by Al-Rowayeh et al. [13] was done on teachers in Kuwait, and the results indicated that 68.5% of teachers reported low back pain, which is similar to our findings in a similar culture.

### 4.2. Effectiveness of Treatment

Sedatives, bed rest, kaiy, and local adhesives were the most common at-home management approaches, with only 9.9% consulting a primary care physician. Moreover, our findings highlight that sedatives are deemed to be the most effective pain-relieving method (34.2%), followed by body piercing (27.2%), exercising (21.3%), and physiotherapy (20%). The current literature corroborates the use of medications to treat this condition, with the guidelines from the American Pain Society and the American College of Physicians recommending paracetamol and NSAIDs as the first-line options for pain relief [17].

Despite the recommended use of medications, the use of sedatives and muscle relaxants to treat non-specific low back pain is advised against. A recent study by Cashin et al. [18] investigated the efficacy, acceptability, and safety of muscle relaxants for low back pain across several populations, totalling 6505 patients. The findings of this study emphasized the considerable uncertainty surrounding the clinical efficacy and safety of sedatives. Moreover, the inadequate management of acute LBP may increase the risk of developing chronic pain in the future [19]. Therefore, seeking medical assistance for managing and treating LBP is strongly recommended.

Although physical therapy and exercise were viewed as the least effective intervention among our participants, the current literature substantiates the therapeutic potential of these management approaches [20,21]. Therefore, early physical therapy for recent onset LBP is advised to achieve the best clinical outcomes. This was proposed by Fritz et al. [22] in a randomized controlled trial that evaluated whether early physical therapy combined with exercise was more effective than the standard of care, including medications, for patients with LBP. The findings noted a significant improvement in quality of life, disability, and pain compared to the usual care for LBP.

Improving the management guidelines for LBP will not only enhance the quality of life of those in sedentary jobs but also represent significant cost savings for the healthcare system and the global economy [23,24,25].

### 4.3. Patient Demographics

Concerning the relationship between patient demographics and the prevalence of back pain, our findings suggest that those below 30 years of age experience a greater prevalence of back pain when compared to those above 50 years of age. These findings are comparable with the current literature. Alnaami et al. [26] investigated the risk factors of LBP amongst workers in southwestern Saudi Arabia, suggesting that those between 20 and 30 years old are significantly more likely to develop LBP. This most likely results from their positions’ longstanding work conditions or highly sedentary behaviour. Parry et al. [27] discussed the significant rise in office work, contributing to sedentary behaviour-associated risk amongst those in the age category above. This is mirrored in our findings, with the data demonstrating an increased prevalence of LBP in participants working in administrative positions when compared to teachers. Although these findings were insignificant, the increased sedentary behaviour in administrative positions may contribute to increased LBP.

Participants who worked every day were significantly more likely to experience back pain than those who worked five or six days a week. Several studies have suggested that working hours spent on repeated activities, such as teaching and administrative jobs, increase the prevalence of LBP [28,29,30]. Koch et al. [31] suggest that these disparities in LBP prevalence arise due to the lack of postural control that teaching and administrative jobs offer. Moreover, our findings suggest that participants that worked for seven hours daily had a significantly increased prevalence of LBP than those that worked four or five hours daily. A study by Watanabe et al. [32] identified the difficulties experienced by workers with LBP. The results highlight that maintaining a seated posture was the most problematic of their working activities. These findings may provide a rationale for the differences in LBP prevalence depending on the daily working hours. Participants who work seven hours daily maintain an uncomfortable posture for longer, exacerbating their LBP.

This study also found that females experienced a greater prevalence of back pain when compared to males, yielding values of 76.3% and 66.1%, respectively. Wáng et al. [33] corroborated these results in a literature review that spanned 98 studies with 772,927 individuals. A collation of the data from each of the included studies deduced an LBP prevalence ratio of females vs. males of 1.310, 1.140, 1.220, and 1.270 across the four age categories, respectively. This highlights that females are disproportionately affected by LBP across all age groups.

### 4.4. Limitations

Despite the strengths of this study, some limitations must be considered. Firstly, we had a small sample size, limiting our results’ external validity. We did not measure types and intensity of back pain; there was only the question of whether a subject had back pain in the format of ‘Yes or No’. Secondly, most (87.9%) of the participants were middle-aged (39.2 years) unmarried men (97.7). Although we can establish an association between participant characteristics, such as working hours, and the prevalence of LBP, we cannot generalize these results and determine causality. Moreover, there is a potential risk of bias, specifically, recall bias and selection bias, in our findings. This may have resulted in an over- or underestimation of our data.

## 5. Conclusions

The current study revealed that about two in three teachers had a history of back pain. This is a notable finding. In addition, there was a significant relationship between workload, including the number of working days and hours, and the prevalence of back pain. Although a considerable number of the included teachers were highly educated, with some teaching experience, they had a poor level of awareness and an incorrect attitude towards pain management. This was evident in their methods of managing their pain, with most of our sample stating that sedatives were their chosen pain-relieving method. The current literature highlights that consulting a medical professional is optimal for LBP; however, seeking medical advice was recorded among few of our participants. Moreover, despite sedatives receiving the highest agreement rate among the teachers as a pain-relieving method, it was agreed on by only one-third of the sample. This indicates a low awareness and poor attitude regarding the role of this medication. Back pain awareness, education regarding the role of sedatives, early physical therapy, and lifestyle modifications should be the routine outreach programs in primary healthcare policies for alleviating low back pain among teachers.

## Figures and Tables

**Figure 1 healthcare-11-00383-f001:**
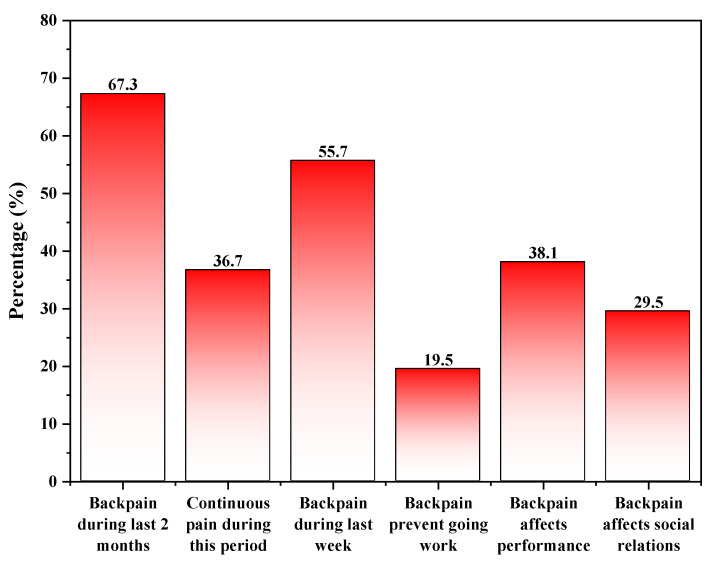
Back pain history among teachers in Asir Region, Saudi Arabia.

**Figure 2 healthcare-11-00383-f002:**
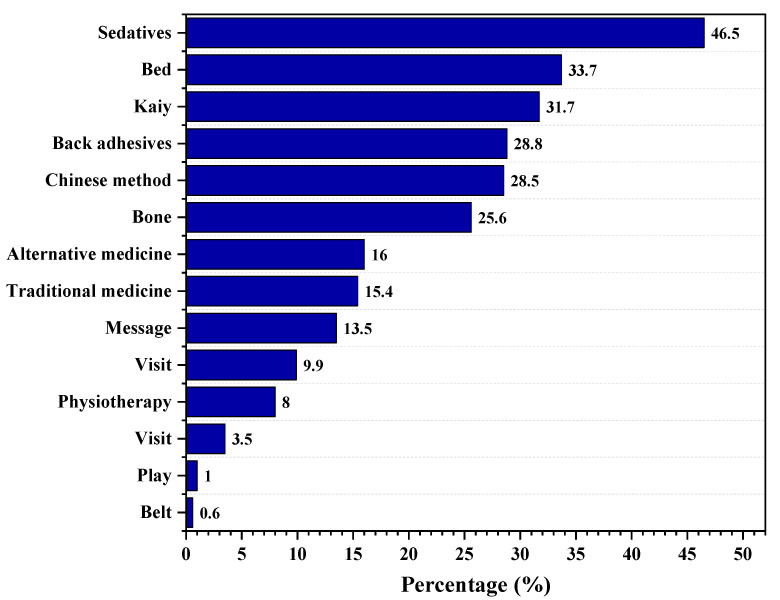
Back pain relief methods, as mentioned by teachers in Asir Region, Saudi Arabia.

**Table 1 healthcare-11-00383-t001:** Socio-demographic characteristics of teachers in Asir Region, Saudi Arabia.

Socio-Demographic Data		No	%
Age in years	<30 years	30	9.6%
	30-	157	50.3%
	40-	108	34.6%
	50–59	17	5.4%
	Mean ± SD		39.2 ± 6.6
Gender	Male	274	87.8%
	Female	38	12.2%
Nationality	Saudi	289	92.6%
	Non-Saudi	23	7.4%
Qualification	Postgraduate	72	29.9%
	Bachelor	130	53.9%
	General Diploma	39	16.2%
Residence	Abha	104	33.3%
	Khamis	132	42.3%
	Barek	63	20.2%
	Other places	13	4.2%
Marital status	Single	304	97.7%
	Divorced	7	2.3%
Smoking	Smoker	256	82.1%
	Non-smoker	56	17.9%

**Table 2 healthcare-11-00383-t002:** Work data of teachers in Asir Region, Saudi Arabia.

Work-Related Data		No	%
Job Nature	Teacher	243	89.7%
	Administrative work	28	10.3%
Working days/week	3.00	1	0.5%
	5.00	199	94.3%
	6.00	2	0.9%
	7.00	9	4.3%
Daily work hours	4.00	3	1.4%
	5.00	45	21.4%
	6.00	51	24.3%
	7.00	111	52.9%

**Table 3 healthcare-11-00383-t003:** Descriptive attitude of teachers regarding back pain relieving methods in Asir Region, Saudi Arabia.

Domain	Item	Disagree		Neutral		Agree	
		No	%	No	%	No	%
Sedatives	Sedatives relieve pain Sedatives useless Afraid of sedatives Sedatives effective as a pain killer	197 127 151 95	63.1% 40.7% 48.4% 30.4%	92 99 108 92	29.5% 31.7% 34.6% 29.5%	23 86 53 125	7.4% 27.6% 17.0% 40.1%
Exercises	Playing exercises relieves pain Playing exercises useless Afraid of playing exercises Exercises effective as a pain killer	217 80 114 162	69.6% 25.6% 36.5% 51.9%	70 115 104 103	22.4% 36.9% 33.3% 33.0%	25 117 94 47	8.0% 37.5% 30.1% 15.1%
Physiotherapy	Physiotherapy relief pain Physiotherapy useless Afraid of physiotherapy Physiotherapy is effective as a pain killer	221 88 111 167	70.8% 28.2% 35.6% 53.5%	70 98 112 115	22.4% 31.4% 35.9% 36.9%	21 126 89 30	6.7% 40.4% 28.5% 9.6%
Body piercing	Body piercing relieves pain Body piercing useless Afraid of body piercing Body piercing is effective as a pain killer	80 89 128 69	25.7% 28.5% 41.2% 22.1%	181 161 146 172	58.2% 51.6% 46.9% 55.1%	50 62 37 71	16.1% 19.9% 11.9% 22.8%

**Table 4 healthcare-11-00383-t004:** Relation between teachers’ characteristics and their history of back pain in Asir Region, Saudi Arabia.

Fact		History of Back Pain				
		Yes				
		No	%	No	%	*p*
Age in years	<30 years 30-	23 102	76.7% 65.0%	7 55	23.3% 35.0%	0.643
	40-	73	67.6%	35	32.4%	
	50–59	12	70.6%	5	29.4%	
Gender	Male	181	66.1%	93	33.9%	0.207
	Female	29	76.3%	9	23.7%	
Job nature	Teacher Administrative work	158 20	65.0% 71.4%	85 8	35.0% 28.6%	0.499
Work days/week	3.00 5.00	1 145	100.0% 72.9%	0 54	0.0% 27.1%	0.049 *
	6.00	1	50.0%	1	50.0%	
	7.00	9	100.0%	0	0.0%	
Daily work hours	4.00 5.00	2 28	66.7% 62.2%	1 17	33.3% 37.8%	0.007 *
	6.00	24	47.1%	27	52.9%	
	7.00	83	74.8%	28	25.2%	

* *p* < 0.05 (significant).

## Data Availability

All data generated or analyzed during this study are available from first author Abdullah Raizah and will be provided on request.

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
