# Peer review of "Treatment Beliefs and Practices towards Low Back Pain among Teachers in Asir Region, Saudi Arabia—A Cross-Sectional Study"

_healthcare, 2023, doi:10.3390/healthcare11030383_

Round 1

Reviewer 1 Report

The text is clearly written and easy to read.

It addresses a public health problem that affects a structuring area for the development of a country, such as education.

The investigation is well elaborated and the methodology is adequate to the objectives of the study. The conclusions are clear and are supported by the data obtained.

The main shortcomings of the paper are related to the limitations of the study. The authors present generalizations that cannot be directly inferred from the results, which are limited to a relatively small area. An example of these aspects can be seen in line 143: “Our results found that 67.3% of teachers in Saudi Arabia experience LBP”. Although they refer to other studies carried out in Saudi Arabia, the statement should be within the context of the text.

Another aspect that should be given some attention are statements that fall short of what would be expected in a scientific article. It would be interesting for the authors to present more concrete solutions and hypotheses for resolving or mitigating the problem. The statements are vague and do not point out possible/plausible paths or solutions. For example: “Based on these findings, the researchers recommend that more effort be made to improve teachers' awareness regarding back pain causes and management methods, but how it can be done. This is especially relevant concerning seeking medical advice. It is also recommended that there is an increased awareness regarding the role of sedatives and lifestyle modification in al-leviating LBP.” More examples, line: 179-180; 183-184; 189-190.

Page 4, figure 2 – twice “Visit”? To correct

Author Response

Dear sir- We were really very happy to see your report, your suggestions and comments had improved our manuscript a lot, we did our level best to change our manuscript as suggested by you. Please find the attached file. 

Once again thanks 

Reviewer 2 Report

Abdullah Raizahand colleagues evaluated pain in teachers in Saudi Arabia. Paper provides an interesting information.

Before acceptance I would suggest some improvements.

1. introduction: authors should provide information on LBP in teachers or similar professions in neighboring countries or those culturally similar to their population. Information from HK is interesting. However, they may not be compared directly with those from Saudi arabia.

2. Results: a. I am missing pain intensity (e.g. VAS score) and type of pain (nociceptive/neuropathic) b. Table 1, martial status: there were no married teachers in the study?

3. Discussion a. results should be compared to the results from similar cultural backgrounds b. limitations. authors provide almost none information for females (12 % of the sample), and none for the married teachers.

Author Response

Dear Reviewer sir- we are highly motivated and thankful after revision reports. It has improved our manuscript a lot. We did our level best to reply your questions and modified our manuscript as suggested. Please find the attach copy.

Round 2

Reviewer 2 Report

Authors improved their manuscript. However, they still didn't improve limitation chapter 4.4. They should clearly state that they did not assess type of pain, intensity measured by VAS and that their results refer mostly to men (women 12 % of sample).

Author Response

Dear Reviewer sir you are absolutly right we did not assess the types and intensity of pain by using VAS, most of our participants were men. These were also our limitations. Modifications are done in the manuscript. Thanks 
